# Generalized Inverse Optimization through Online Learning

**Chaosheng Dong**
Department of Industrial Engineering
University of Pittsburgh
chaosheng@pitt.edu

**Yiran Chen**
Department of Electrical and Computer Engineering
Duke University
yiran.chen@duke.edu

**Bo Zeng**
Department of Industrial Engineering
University of Pittsburgh
bzeng@pitt.edu

## Abstract

Inverse optimization is a powerful paradigm for learning preferences and restrictions that explain the behavior of a decision maker, based on a set of external signal and the corresponding decision pairs. However, most inverse optimization algorithms are designed specifically in batch setting, where all the data is available in advance. As a consequence, there has been rare use of these methods in an online setting suitable for real-time applications. In this paper, we propose a general framework for inverse optimization through online learning. Specifically, we develop an online learning algorithm that uses an implicit update rule which can handle noisy data. Moreover, under additional regularity assumptions in terms of the data and the model, we prove that our algorithm converges at a rate of $\mathcal{O}(1/\sqrt{T})$ and is statistically consistent. In our experiments, we show the online learning approach can learn the parameters with great accuracy and is very robust to noises, and achieves a dramatic improvement in computational efficacy over the batch learning approach.

## 1 Introduction

Possessing the ability to elicit customers' preferences and restrictions (PR) is crucial to the success for an organization in designing and providing services or products. Nevertheless, as in most scenarios, one can only observe their decisions or behaviors corresponding to external signals, while cannot directly access their decision making schemes. Indeed, decision makers probably do not have exact information regarding their own decision making process [1]. To bridge that discrepancy, inverse optimization has been proposed and received significant research attention, which is to infer or learn the missing information of the underlying decision models from observed data, assuming that human decision makers are rationally making decisions [2, 3, 4, 5, 1, 6, 7, 8, 9, 10, 11]. Nowadays, extending from its initial form that only considers a single observation [2, 3, 4, 5] with clean data, inverse optimization has been further developed and applied to handle more realistic cases that have many observations with noisy data [1, 6, 7, 9, 10, 11].

Despite of these remarkable achievements, traditional inverse optimization (typically in batch setting) has not proven fully applicable for supporting recent attempts in AI to automate the elicitation of human decision maker's PR in real time. Consider, for example, recommender systems (RSs) used by online retailers to increase product sales. The RSs first elicit one customer's PR from the

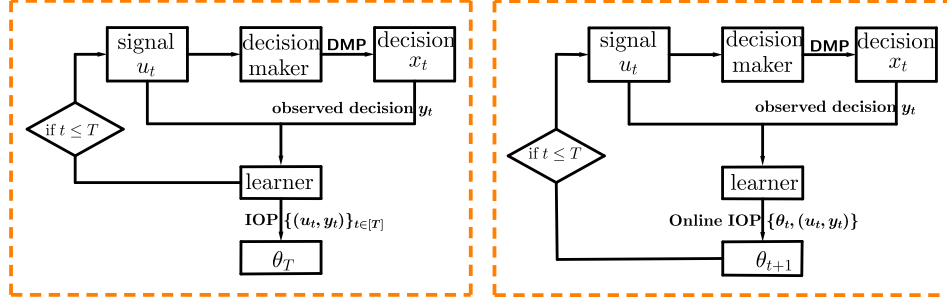

Figure 1: An overview of inverse optimization through batch learning versus through online learning. **Left:** Framework of inverse optimization in batch setting. **Right:** Framework of the generalized inverse optimization in online setting proposed in our paper.

historical sequence of her purchasing behaviors, and then make predictions about her future shopping actions. Indeed, building RSs for online retailers is challenging because of the sparsity issue. Given the large amount of products available, customer's shopping vector, each element of which represents the quantity of one product purchased, is highly sparse. Moreover, the shift of the customer's shopping behavior along with the external signal (e.g. price, season) aggravates the sparsity issue. Therefore, it is particularly important for RSs to have access to large data sets to perform accurate elicitation [12]. Considering the complexity of the inverse optimization problem (IOP), it will be extremely difficult and time consuming to extract user's PR from large, noisy data sets using conventional techniques. Thus, incorporating traditional inverse optimization into RSs is impractical for real time elicitation of user's PR.

To automate the elicitation of human decision maker's PR, we aim to unlock the potential of inverse optimization through online learning in this paper. Specifically, we formulate such learning problem as an IOP considering noisy data, and develop an online learning algorithm to derive unknown parameters occurring in either the objective function or constraints. At the heart of our algorithm is taking inverse optimization with a single observation as a subroutine to define an implicit update rule. Through such an implicit rule, our algorithm can rapidly incorporate sequentially arrived observations into this model, without keeping them in memory. Indeed, we provide a general mechanism for the incremental elicitation, revision and reuse of the inference about decision maker's PR.

**Related work** Our work is most related to the subject of inverse optimization with multiple observations. The goal is to find an objective function or constraints that explains the observations well. This subject actually carries the data-driven concept and becomes more applicable as large amounts of data are generated and become readily available, especially those from digital devices and online transactions. Solution methods in batch setting for such type of IOP include convex optimization approach [1, 13, 10] and non-convex optimization approach [7]. The former approach often yields incorrect inferences of the parameters [7] while the later approach is known to lead to intractable programs to solve [10]. In contrast, we do inverse optimization in online setting, and the proposed online learning algorithm significantly accelerate the learning process with performance guarantees, allowing us to deal with more realistic and complex PR elicitation problems.

Also related to our work is [6], which develops an online learning method to infer the utility function from sequentially arrived observations. They prove a different regret bound for that method under certain conditions, and demonstrate its applicability to handle both continuous and discrete decisions. However, their approach is only possible when the utility function is linear and the data is assumed to be noiseless. Differently, our approach does not make any such assumption and only requires the convexity of the underlying decision making problem. Besides the regret bound, we also show the statistical consistency of our algorithm by applying both the consistency result proven in [7] and the regret bound provided in this paper, which guarantees that our algorithm will asymptotically achieves the best prediction error permitted by the inverse model we consider.

**Our contributions** To the best of authors' knowledge, we propose the first general framework for eliciting decision maker's PR using inverse optimization through online learning. This framework can learn general convex utility functions and constraints with observed (signal, noisy decision) pairs. In Figure 1, we provide the comparison of inverse optimization through batch learning versus through online learning. Moreover, we prove that the online learning algorithm, which adopts

an implicit update rule, has a $\mathcal{O}(\sqrt{T})$ regret under certain regularity conditions. In addition, this algorithm is statistically consistent when the data satisfies some rather common conditions, which guarantees that our algorithm will asymptotically achieves the best prediction error permitted by the inverse model we consider. Numerical results show that our algorithm can learn the parameters with great accuracy, is robust to noises even if some assumptions do not hold, and achieves a dramatic improvement over the batch learning approach on computational efficacy.

## 2 Problem setting

### 2.1 Decision making problem

We consider a family of parameterized decision making problems, in which $\mathbf{x} \in \mathbb{R}^n$ is the decision variable, $u \in \mathcal{U} \subseteq \mathbb{R}^m$ is the external signal, and $\theta \in \Theta \subseteq \mathbb{R}^p$ is the parameter.

$$\min_{\mathbf{x} \in \mathbb{R}^n} \quad f(\mathbf{x}, u, \theta) \qquad\qquad \text{DMP}$$
$$s.t. \quad \mathbf{g}(\mathbf{x}, u, \theta) \leq \mathbf{0},$$

where $f : \mathbb{R}^n \times \mathbb{R}^m \times \mathbb{R}^p \mapsto \mathbb{R}$ is a real-valued function, and $\mathbf{g} : \mathbb{R}^n \times \mathbb{R}^m \times \mathbb{R}^p \mapsto \mathbb{R}^q$ is a vector-valued function. We denote $X(u, \theta) = \{x \in \mathbb{R}^n : \mathbf{g}(\mathbf{x}, u, \theta) \leq \mathbf{0}\}$ the feasible region of DMP. We let $S(u, \theta) = \arg\min\{f(\mathbf{x}, u, \theta) : x \in X(u, \theta)\}$ be the optimal solution set of DMP.

### 2.2 Inverse optimization and online setting

Consider a learner who monitors the signal $u \in \mathcal{U}$ and the decision maker' decision $\mathbf{x} \in X(u, \theta)$ in response to $u$. We assume that the learner does not know the decision maker's utility function or constraints in DMP. Since the observed decision might carry measurement error or is generated with a bounded rationality of the decision maker, i.e., being suboptimal, we denote $\mathbf{y}$ the observed noisy decision for $u \in \mathcal{U}$. Note that $\mathbf{y}$ does not necessarily belong to $X(u, \theta)$, i.e., it might be infeasible with respect to $X(u, \theta)$. Throughout the paper, we assume that the (signal,noisy decision) pair $(u, \mathbf{y})$ is distributed according to some unknown distribution $\mathbb{P}$ supported on $\{(u, \mathbf{y}) : u \in \mathcal{U}, \mathbf{y} \in \mathcal{Y}\}$.

In our inverse optimization model, the learner aims to learn the decision maker's objective function or constraints from (signal, noisy decision) pairs. More precisely, the goal of the learner is to estimate the parameter $\theta$ of the DMP. In our online setting, the (signal, noisy decision) pair become available to the learner one by one. Hence, the learning algorithm produces a sequence of hypotheses $(\theta_1, \ldots, \theta_{T+1})$. Here, $T$ is the total number of rounds, and $\theta_1$ is an arbitrary initial hypothesis and $\theta_t$ for $t \geq 2$ is the hypothesis chosen after observing the $(t-1)$th (signal,noisy decision) pair. Let $l(\mathbf{y}_t, u_t, \theta_t)$ denote the loss the learning algorithm suffers when it tries to predict the $t$th decision given $u_t$ based on $\{(u_1, \mathbf{y}_1), \cdots, (u_{t-1}, \mathbf{y}_{t-1})\}$. The goal of the learner is to minimize the regret, which is the cumulative loss $\sum_{t \in [T]} l(\mathbf{y}_t, u_t, \theta_t)$ against the possible loss when the whole batch of (signal,noisy decision) pairs are available. Formally, the regret is defined as

$$R_T = \sum_{t \in [T]} l(\mathbf{y}_t, u_t, \theta_t) - \min_{\theta \in \Theta} \sum_{t \in [T]} l(\mathbf{y}_t, u_t, \theta).$$

In the following, we make a few assumptions to simplify our understanding, which are actually mild and frequently appear in the inverse optimization literature [1, 13, 10, 7].

**Assumption 2.1.** Set $\Theta$ is a convex compact set. There exists $D > 0$ such that $\|\theta\|_2 \leq D$ for all $\theta \in \Theta$. In addition, for each $u \in \mathcal{U}, \theta \in \Theta$, both $\mathbf{f}(\mathbf{x}, u, \theta)$ and $\mathbf{g}(\mathbf{x}, u, \theta)$ are convex in $\mathbf{x}$.

## 3 Learning the parameters

### 3.1 The loss function

Different loss functions that capture the mismatch between predictions and observations have been used in the inverse optimization literature. In particular, the (squared) distance between the observed decision and the predicted decision enjoys a direct physical meaning, and thus is most widely used [14, 15, 16, 7]. Hence, we take the (squared) distance as our loss function in this paper.In batch

setting, statistical properties of inverse optimization with such a loss function have been analyzed extensively in [7]. In this paper, we focus on exploring the performance of the online setting.

Given a (signal,noisy decision) pair $(u, \mathbf{y})$ and a hypothesis $\theta$, we define the following loss function as the minimum (squared) distance between $\mathbf{y}$ and the optimal solution set $S(u, \theta)$.

$$l(\mathbf{y}, u, \theta) = \min_{\mathbf{x} \in S(u,\theta)} \|\mathbf{y} - \mathbf{x}\|_2^2. \qquad \text{Loss Function}$$

### 3.2 Online implicit updates

Once receiving the $t$th (signal,noisy decision) pair $(u_t, \mathbf{y}_t)$, $\theta_{t+1}$ can be obtained by solving the following optimization problem:

$$\theta_{t+1} = \arg \min_{\theta \in \Theta} \quad \tfrac{1}{2}\|\theta - \theta_t\|_2^2 + \eta_t l(\mathbf{y}_t, u_t, \theta), \qquad (1)$$

where $\eta_t$ is the learning rate in round $t$, and $l(\mathbf{y}_t, u_t, \theta)$ is defined in (Loss Function).

The updating rule (1) seeks to balance the tradeoff between "conservativeness" and correctiveness", where the first term characterizes how conservative we are to maintain the current estimation, and the second term indicates how corrective we would like to modify with the new estimation. As there is no closed form for $\theta_{t+1}$ in general, we call (1) an implicit update rule [17, 18].

To solve (1), we can replace $\mathbf{x} \in S(u, \theta)$ by KKT conditions (or other optimality conditions) of the DMP, and get a mixed integer nonlinear program. Consider, for example, a decision making problem that is a quadratic optimization problem. Namely, the DMP has the following form:

$$\min_{\mathbf{x} \in \mathbb{R}^n} \quad \tfrac{1}{2}\mathbf{x}^T Q \mathbf{x} + \mathbf{c}^T \mathbf{x}$$
$$s.t. \quad A\mathbf{x} \geq \mathbf{b}. \qquad \text{QP}$$

Suppose that $\mathbf{b}$ changes over time $t$. That is, $\mathbf{b}$ is the external signal for QP and equals to $\mathbf{b}_t$ at time $t$. If we seek to learn $\mathbf{c}$, the optimal solution set for QP can be characterized by KKT conditions as $S(\mathbf{b}_t) = \{\mathbf{x} : A\mathbf{x} \geq \mathbf{b}_t, \ \mathbf{u} \in \mathbb{R}_+^m, \ \mathbf{u}^T(A\mathbf{x} - \mathbf{b}_t) = 0, \ Q\mathbf{x} + \mathbf{c} - A^T\mathbf{u} = 0\}$. Here, $\mathbf{u}$ is the dual variable for the constraints. Then, the single level reformulation of the update rule by solving (1) is

$$\min_{\mathbf{c} \in \Theta} \quad \tfrac{1}{2}\|\mathbf{c} - \mathbf{c}_t\|_2^2 + \eta_t \|\mathbf{y}_t - \mathbf{x}\|_2^2$$
$$s.t. \quad A\mathbf{x} \geq \mathbf{b}_t,$$
$$\mathbf{u} \leq M\mathbf{z},$$
$$A\mathbf{x} - \mathbf{b}_t \leq M(1 - \mathbf{z}), \qquad \text{IQP}$$
$$Q\mathbf{x} + \mathbf{c} - A^T\mathbf{u} = 0,$$
$$\mathbf{c} \in \mathbb{R}^m, \ \mathbf{x} \in \mathbb{R}^n, \ \mathbf{u} \in \mathbb{R}_+^m, \ \mathbf{z} \in \{0,1\}^m,$$

where $\mathbf{z}$ is the binary variable used to linearize KKT conditions, and $M$ is an appropriate number used to bound the dual variable $\mathbf{u}$ and $A\mathbf{x} - \mathbf{b}_t$. Clearly, IQP is a mixed integer second order conic program (MISOCP). More examples are given in the supplementary material.

Our application of the implicit updates to learn the parameter of DMP proceeds in Algorithm 1.

**Remark 3.1.** $(i)$ In Algorithm 1, we let $\theta_{t+1} = \theta_t$ if the prediction error $l(\mathbf{y}_t, u_t, \theta_t)$ is zero. But in practice, we can set a threshold $\epsilon > 0$ and let $\theta_{t+1} = \theta_t$ once $l(\mathbf{y}_t, u_t, \theta_t) < \epsilon$. $(ii)$ Normalization of $\theta_{t+1}$ is needed in some situations, which eliminates the impact of trivial solutions. $(iii)$ **Mini-batches** One technique to enhance online learning is to consider multiple observations per update. In our framework, this means that computing $\theta_{t+1}$ using $|N_t| > 1$ noisy decisions in (1).

**Remark 3.2.** To obtain a strong initialization of $\theta$ in Algorithm 1, we can incorporate an idea in [1], which imputes a convex objective function by minimizing the residuals of KKT conditions incurred by the noisy data. Assume we have a historical data set $\widetilde{T}$, which may be of poor qualities for the current learning. This leads to the following initialization problem:

$$\min_{\theta \in \Theta} \quad \frac{1}{|\widetilde{T}|} \sum_{t \in [\widetilde{T}]} \left( r_c^t + r_s^t \right)$$
$$s.t. \quad |\mathbf{u}_t^T \mathbf{g}(\mathbf{y}_t, u_t, \theta)| \leq r_c^t, \qquad\qquad \forall t \in \widetilde{T},$$
$$\|\nabla f(\mathbf{y}_t, u_t, \theta) + \nabla \mathbf{u}_t^T \mathbf{g}(\mathbf{y}_t, u_t, \theta)\|_2 \leq r_s^t, \quad \forall t \in \widetilde{T}, \qquad (2)$$
$$\mathbf{u}_t \in \mathbb{R}_+^m, \ r_c^t \in \mathbb{R}_+, \ r_s^t \in \mathbb{R}_+, \qquad\qquad \forall t \in \widetilde{T},$$

---

**Algorithm 1** Implicit Online Learning for Generalized Inverse Optimization

---
1: **Input:** (signal,noisy decision) pairs $\{(u_t, \mathbf{y}_t)\}_{t \in [T]}$
2: **Initialization:** $\theta_1$ could be an arbitrary hypothesis of the parameter.
3: **for** $t = 1$ to $T$ **do**
4:     receive $(u_t, \mathbf{y}_t)$
5:     suffer loss $l(\mathbf{y}_t, u_t, \theta_t)$
6:     **if** $l(\mathbf{y}_t, u_t, \theta_t) = 0$ **then**
7:         $\theta_{t+1} \leftarrow \theta_t$
8:     **else**
9:         set learning rate $\eta_t \propto 1/\sqrt{t}$
10:         update $\theta_{t+1} = \arg\min_{\theta \in \Theta} \frac{1}{2}\|\theta - \theta_t\|_2^2 + \eta_t l(\mathbf{y}_t, u_t, \theta)$ (solve (1))
11:     **end if**
12: **end for**

---

where $r_c^t$ and $r_s^t$ are residuals corresponding to the complementary slackness and stationarity in KKT conditions for the $t$-th noisy decision $\mathbf{y}_t$, and $\mathbf{u}_t$ is the dual variable corresponding to the constraints in DMP. Note that (2) is a convex program. It can be solved quite efficiently compared to solving the inverse optimization problem in batch setting [7]. Other initialization approaches using similar ideas e.g., computing a variational inequality based approximation of inverse model [13], can also be incorporated into our algorithm.

### 3.3 Theoretical analysis

Note that the implicit online learning algorithm is generally applicable to learn the parameter of any convex DMP. In this section, we prove that the average regret $R_T/T$ converges at a rate of $\mathcal{O}(1/\sqrt{T})$ under certain regularity conditions. Furthermore, we will show that the proposed algorithm is statistically consistent when the data satisfies some common regularity conditions. We begin by introducing a few assumptions that are rather common in literature [1, 13, 10, 7].

**Assumption 3.1. (a)** For each $u \in \mathcal{U}$ and $\theta \in \Theta$, $X(u, \theta)$ is closed, and has a nonempty relative interior. $X(u, \theta)$ is also uniformly bounded. That is, there exists $B > 0$ such that $\|\mathbf{x}\|_2 \leq B$ for all $\mathbf{x} \in X(u, \theta)$.

**(b)** $f(\mathbf{x}, u, \theta)$ is $\lambda$-strongly convex in $\mathbf{x}$ on $\mathcal{Y}$ for fixed $u \in \mathcal{U}$ and $\theta \in \Theta$. That is, $\forall \mathbf{x}, \mathbf{y} \in \mathcal{Y}$,

$$\left(\nabla f(\mathbf{y}, u, \theta) - \nabla f(\mathbf{x}, u, \theta)\right)^T (\mathbf{y} - \mathbf{x}) \geq \lambda\|\mathbf{x} - \mathbf{y}\|_2^2.$$

**Remark 3.3.** For strongly convex program, there exists only one optimal solution. Therefore, Assumption 3.1.(b) ensures that $S(u, \theta)$ is a single-valued set for each $u \in \mathcal{U}$. However, $S(u, \theta)$ might be multivalued for general convex DMP for fixed $u$. Consider, for example, $\min_{x_1, x_2}\{x_1 + x_2 : x_1 + x_2 \geq 1\}$. Note that all points on line $x_1 + x_2 = 1$ are optimal. Indeed, we find such case is quite common when there are many variables and constraints. Actually, it is one of the major challenges when learning parameters of a function that's not strongly convex using inverse optimization.

For convenience of analysis, we assume below that we seek to learn the objective function while constraints are known. Then, the performance of Algorithm 1 also depends on how the change of $\theta$ affects the objective values. For $\forall \mathbf{x} \in \mathcal{Y}, \forall u \in \mathcal{U}, \forall \theta_1, \theta_2 \in \Theta$, we consider the difference function

$$h(\mathbf{x}, u, \theta_1, \theta_2) = f(\mathbf{x}, u, \theta_1) - f(\mathbf{x}, u, \theta_2). \tag{3}$$

**Assumption 3.2.** $\exists \kappa > 0, \forall u \in \mathcal{U}, \forall \theta_1, \theta_2 \in \Theta$, $h(\cdot, u, \theta_1, \theta_2)$ is Lipschitz continuous on $\mathcal{Y}$:

$$|h(\mathbf{x}, u, \theta_1, \theta_2) - h(\mathbf{y}, u, \theta_1, \theta_2)| \leq \kappa\|\theta_1 - \theta_2\|_2\|\mathbf{x} - \mathbf{y}\|_2, \forall \mathbf{x}, \mathbf{y} \in \mathcal{Y}.$$

Basically, this assumption says that the objectives functions will not change very much when either the parameter $\theta$ or the variable $\mathbf{x}$ is perturbed. It actually holds in many common situations, including the linear program and quadratic program.

**Lemma 3.1.** Under Assumptions 2.1 - 3.2, the loss function $l(\mathbf{y}, u, \theta)$ is uniformly $\frac{4(B+R)\kappa}{\lambda}$-Lipschitz continuous in $\theta$. That is, $\forall \mathbf{y} \in \mathcal{Y}, \forall u \in \mathcal{U}, \forall \theta_1, \theta_2 \in \Theta$, we have

$$|l(\mathbf{y}, u, \theta_1) - l(\mathbf{y}, u, \theta_2)| \leq \frac{4(B+R)\kappa}{\lambda}\|\theta_1 - \theta_2\|_2.$$

The establishment of Lemma 3.1 is based on the key observation that the perturbation of $S(u, \theta)$ due to $\theta$ is bounded by the perturbation of $\theta$ through applying Proposition 6.1 in [19]. Details of the proof are given in the supplementary material.

**Remark 3.4.** When we seek to learn the constraints or jointly learn the constraints and objective function, similar result can be established by applying Proposition 4.47 in [20] while restricting not only the Lipschitz continuity of the difference function in (3), but also the Lipschitz continuity of the distance between the feasible sets $X(u, \theta_1)$ and $X(u, \theta_2)$ (see Remark 4.40 in [20]).

**Assumption 3.3.** For the DMP, $\forall \mathbf{y} \in \mathcal{Y}, \forall u \in \mathcal{U}, \forall \theta_1, \theta_2 \in \Theta, \forall \alpha, \beta \geq 0$ s.t. $\alpha + \beta = 1$, we have

$$\|\alpha S(u, \theta_1) + \beta S(u, \theta_2) - S(u, \alpha\theta_1 + \beta\theta_2)\|_2 \leq \alpha\beta\|S(u, \theta_1) - S(u, \theta_2)\|_2/(2(B+R)).$$

Essentially, this assumption indicates that the distance between $S(u, \alpha\theta_1 + \beta\theta_2)$ and the convex combination of $S(u, \theta_1)$ and $S(u, \theta_2)$ shall be small when $S(u, \theta_1)$ and $S(u, \theta_2)$ are close. An example is provided in the supplementary material to show that this assumption can be satisfied. Yet, we note that it probably is restrictive and hard to verify in general.

Let $\theta^*$ be an optimal inference to $\min_{\theta \in \Theta} \frac{1}{T} \sum_{t \in [T]} l(\mathbf{y}_t, \theta)$, i.e., an inference derived with the whole batch of observations available. Then, the following theorem asserts that $R_T = \sum_{t \in [T]} (l(\mathbf{y}_t, \theta_t) - l(\mathbf{y}_t, \theta^*))$ of the implicit online learning algorithm is of $\mathcal{O}(\sqrt{T})$.

**Theorem 3.2** (Regret bound). Suppose Assumptions 2.1 - 3.3 hold. Then, choosing $\eta_t = \frac{D\lambda}{2\sqrt{2}(B+R)\kappa} \frac{1}{\sqrt{t}}$, we have

$$R_T \leq \frac{8\sqrt{2}(B+R)D\kappa}{\lambda} \sqrt{T}.$$

**Remark 3.5.** We establish of the above regret bound by extending Theorem 3.2. in [18]. Our extension involves several critical and complicated analyses for the structure of the optimal solution set $S(u, \theta)$ as well as the loss function, which is essential to our theoretical understanding. Moreover, we relax the requirement of smoothness of loss function in that theorem to Lipschitz continuity through a similar argument in Lemma 1 of [21] and [22].

By applying both Theorem 3 in [7] and the regret bound proved in Theorem 3.2, we show the risk consistency of the online learning algorithm in the sense that the average cumulative loss converges in probability to the true risk in the batch setting.

**Theorem 3.3** (Risk consistency). Let $\theta^0 = \arg\min_{\theta \in \Theta}\{\mathbb{E}[l(\mathbf{y}, u, \theta)]\}$ be the optimal solution that minimizes the true risk in batch setting. Suppose the conditions in Theorem 3.2 hold. If $\mathbb{E}[\mathbf{y}^2] < \infty$, then choosing $\eta_t = \frac{D\lambda}{2\sqrt{2}(B+R)\kappa} \frac{1}{\sqrt{t}}$, we have

$$\frac{1}{T} \sum_{t \in [T]} l(\mathbf{y}_t, u_t, \theta_t) \xrightarrow{p} \mathbb{E}[l(\mathbf{y}, u, \theta^0)].$$

**Corollary 3.3.1.** Suppose that the true parameter $\theta_{true} \in \Theta$, and $\mathbf{y} = \mathbf{x} + \epsilon$, where $\mathbf{x} \in S(u, \theta_{true})$ for some $u \in \mathcal{U}$, $\mathbb{E}[\epsilon] = 0, \mathbb{E}[\epsilon^T \epsilon] < \infty$, and $u, \mathbf{x}$ are independent of $\epsilon$. Let the conditions in Theorem 3.2 hold. Then choosing $\eta_t = \frac{D\lambda}{2\sqrt{2}(B+R)\kappa} \frac{1}{\sqrt{t}}$, we have

$$\frac{1}{T} \sum_{t \in [T]} l(\mathbf{y}_t, u_t, \theta_t) \xrightarrow{p} \mathbb{E}[\epsilon^T \epsilon].$$

**Remark 3.6.** $(i)$ Theorem 3.3 guarantees that the online learning algorithm proposed in this paper will asymptotically achieves the best prediction error permitted by the inverse model we consider. $(ii)$ Corollary 3.3.1 suggests that the prediction error is inevitable as long as the data carries noise. This prediction error, however, will be caused merely by the noisiness of the data in the long run.

## 4 Applications to learning problems in IOP

In this section, we will provide sketches of representative applications for inferring objective functions and constraints using the proposed online learning algorithm. Our preliminary experiments have been run on Bridges system at the Pittsburgh Supercomputing Center (PSC) [23]. The mixed integer second order conic programs, which are derived from using KKT conditions in (1), are solved by Gurobi. All the algorithms are programmed with Julia [24].

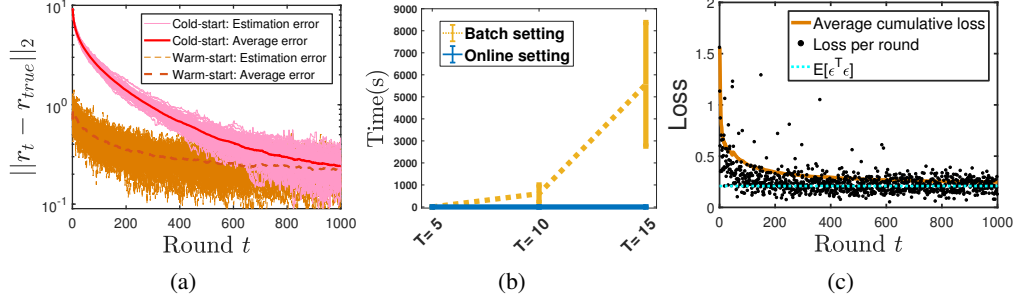

(a)                                      (b)                                      (c)

Figure 2: Learning the utility function over $T = 1000$ rounds. (a) We run 100 repetitions of the experiments using Algorithm 1 with two settings. **Cold-start** means that we initialize $\mathbf{r}$ as a vector of zeros. **Warm-start** means that we initialize $\mathbf{r}$ by solving (2) with 1000 (price,noisy decision) pairs. We plot the estimation errors over round $t$ in pink and brown for all the 100 repetitions, respectively. We also plot the average estimation errors of the 100 repetitions in red line and dashed brown line, respectively. (b) The dotted brown line is the error bar plot of the average running time over 10 repetitions in batch setting. The blue line is the error bar plot of the average running time over 100 repetitions in online setting. Here, the error bar is [mean-std, mean+std]. (c) We randomly pick one repetition. The loss over round is indicated by the dot. The average cumulative loss is indicated by the line. The dotted line indicates the variance of the noise. Here, $\mathbf{E}[\epsilon^T \epsilon] = 0.2083$.

## 4.1 Learning consumer behavior

We now study the consumer's behavior problem in a market with $n$ products. The prices for the products are denoted by $\mathbf{p}_t \in \mathbb{R}^n_+$ which varies over time $t \in [T]$. We assume throughout that the consumer has a rational preference relation, and we take $u$ to be the utility function representing these preferences. The consumer's decision making problem of choosing her most preferred consumption bundle $\mathbf{x}$ given the price vector $\mathbf{p}_t$ and budget $b$ can be stated as the following utility maximization problem (UMP) [25].

$$\max_{\mathbf{x} \in \mathbb{R}^n_+} \quad u(\mathbf{x})$$
$$s.t. \quad \mathbf{p}_t^T \mathbf{x} \le b, \qquad \qquad \text{UMP}$$

where $\mathbf{p}_t^T \mathbf{x} \le b$ is the budget constraint at time $t$.

For this application, we will consider a concave quadratic representation for $u(\mathbf{x})$. That is, $u(\mathbf{x}) = \frac{1}{2}\mathbf{x}^T Q \mathbf{x} + \mathbf{r}^T \mathbf{x}$, where $Q \in \mathbf{S}^n_-$ (the set of symmetric negative semidefinite matrices), $\mathbf{r} \in \mathbb{R}^n$.

We consider a problem with $n = 10$ products, and the budget $b = 40$. $Q$ and $\mathbf{r}$ are randomly generated and are given in the supplementary material. Suppose prices are changing in $T$ rounds. In each round, the learner would receive one (price,noisy decision) pair $(\mathbf{p}_t, \mathbf{y}_t)$. Her goal is to learn the utility function or budget of the consumer. The (price,noisy decision) pair in each round is generated as follows. In round $t$, we generate the prices from a uniform distribution, i.e. $p_i^t \sim U[p_{min}, p_{max}]$, with $p_{min} = 5$ and $p_{max} = 25$. Then, we solve UMP and get the optimal decision $\mathbf{x}_t$. Next, the noisy decision $\mathbf{y}_t$ is obtained by corrupting $\mathbf{x}_t$ with noise that has a jointly uniform distribution with support $[-0.25, 0.25]^2$. Namely, $\mathbf{y}_t = \mathbf{x}_t + \epsilon_t$, where each element of $\epsilon_t \sim U(-0.25, 0.25)$.

**Learning the utility function** In the first set of experiments, the learner seeks to learn $\mathbf{r}$ given $\{(\mathbf{p}_t, \mathbf{y}_t)\}_{t \in [T]}$ that arrives sequentially in $T = 1000$ rounds. We assume that $\mathbf{r}$ is within $[0, 5]^{10}$. The learning rate is set to $\eta_t = 5/\sqrt{t}$. Then, we implement Algorithm 1 with two settings. We report our results in Figure 2. As can be seen in Figure 2a, solving the initialization problem provides quite good initialized estimations of $\mathbf{r}$, and Algorithm 1 with Warm-start converges faster than that with Cold-start. Note that (2) is a convex program and the time to solve it is negligible in Algorithm 1. Thus, the running times with and without Warm-start are roughly the same. This suggests that one might prefer to use Algorithm 1 with Warm-start if she wants to get a relatively good estimation of the parameters in few iterations. However, as shown in the figure, both settings would return very similar estimations on $\mathbf{r}$ in the long run. To keep consistency, we would use Algorithm 1 with Cold-start in the remaining experiments. We can also see that estimation errors over rounds for different repetitions concentrate around the average, indicating that our algorithm is pretty robust to noises. Moreover, Figure 2b shows that inverse optimization in online setting is drastically faster

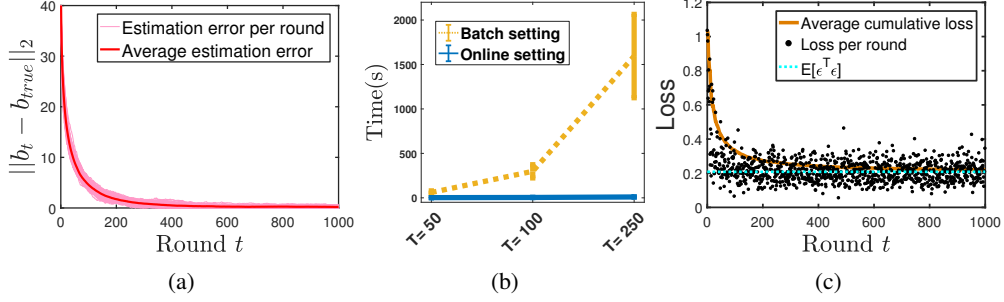

Figure 3: Learning the budget over $T = 1000$ rounds. (a) We run 100 repetitions of the experiments. We plot the estimation error over round $t$ for all the 100 repetitions in pink. We also plot the average estimation error of the 100 repetitions in red. (b) The dotted brown line is the error bar plot of the average running time over 10 repetitions in batch setting. The blue line is the error bar plot of the average running time over 100 repetitions in online setting. (c) We randomly pick one repetition. The loss over round is indicated by the dot. The average cumulative loss is indicated by the line. The dotted line is the reference line indicating the variance of the noise. Here, $\mathbf{E}[\epsilon^T \epsilon] = 0.2083$.

than in batch setting. This also suggests that windowing approach for inverse optimization might be practically infeasible since it fails even with a small subset of data, such as window size equals to 10. We then randomly pick one repetition and plot the loss over round and the average cumulative loss in Figure 2c. We see clearly that the average cumulative loss asymptotically converges to the variance of the noise. This makes sense because the loss merely reflects the noise in the data when the estimation converges to the true value as stated in Remark 3.6.

**Learning the budget**   In the second set of experiments, the learner seeks to learn the budget $b$ in $T = 1000$ rounds. We assume that $b$ is within $[0, 100]$. The learning rate is set to $\eta_t = 100/\sqrt{t}$. Then, we apply Algorithm 1 with Cold-start. We show the results in Figure 3. All the analysis for the results in learning the utility function apply here. One thing to emphasize is that learning the budget is much faster than learning the utility function, as shown in Figure 2b and 3b. The main reason is that the budget $\mathbf{b}$ is a one dimensional vector, while the utility vector $\mathbf{r}$ is a ten dimensional vector, making it drastically more complex to solve (1).

## 4.2   Learning the transportation cost

We now consider the transshipment network $G = (V_s \cup V_d, E)$, where nodes $V_s$ are producers and the remaining nodes $V_d$ are consumers. The production level is $y_v$ for node $v \in V_s$, and has a maximum capacity of $w_v$. The demand level is $d_v^t$ for node $v \in V_s$ and varies over time $t \in [T]$. We assume that producing $y_v$ incurs a cost of $C^v(y_v)$ for node $v \in V_s$; furthermore, we also assume that there is a transportation cost $c_e x_e$ associated with edge $e \in E$, and the flow $x_e$ has a maximum capacity of $u_e$. The transshipment problem can be formulated in the following:

$$
\begin{aligned}
\min \quad & \sum_{v \in V_s} C^v(y_v) + \sum_{e \in E} c_e x_e \\
s.t. \quad & \sum_{e \in \delta^+(v)} x_e - \sum_{e \in \delta^-(v)} x_e = y_v, \quad \forall v \in V_s, \\
& \sum_{e \in \delta^+(v)} x_e - \sum_{e \in \delta^-(v)} x_e = d_v^t, \quad \forall v \in V_d, \\
& 0 \leq x_e \leq u_e, \ \ 0 \leq y_v \leq w_v, \quad \forall e \in E, \forall v \in V_s,
\end{aligned}
\qquad \text{TP}
$$

where we want to learn the transportation cost $c_e$ for $e \in E$. For this application, we will consider a convex quadratic cost for $C^v(y_v)$. That is, $C^v(y_v) = \frac{1}{2}\lambda_v y_v^2$, where $\lambda_v \geq 0$.

We create instances of the problem based on the network in Figure 4a. $\lambda_1, \lambda_2, \{u_e\}_{e \in E}, \{w_v\}_{v \in V_s}$ and the randomly generated $\{c_e\}_{e \in E}$ are given in supplementary material. In each round, the learner would receive the demands $\{d_v^t\}_{v \in V_d}$, the production levels $\{y_v\}_{v \in V_s}$ and the flows $\{x_e\}_{e \in E}$, where the later two are corrupted by noises. In round $t$, we generate the $d_v^t$ for $v \in V_d$ from a uniform distribution, i.e. $d_v^t \sim U[-1.25, 0]$. Then, we solve TP and get the optimal production levels and flows. Next, the noisy production levels and flows are obtained by corrupting the optimal ones with noise that has a jointly uniform distribution with support $[-0.25, 0.25]^8$.

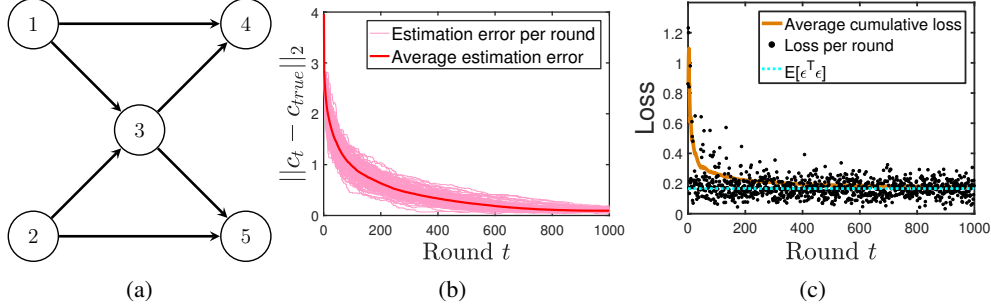

(a)                     (b)                     (c)

Figure 4: Learning the transportation cost over $T = 1000$ rounds. (a) We plot the five-node network in our experiment. (b) Denote $c \in \mathbb{R}^{|E|}$ the vector of transportation costs. We run 100 repetitions of the experiments. We plot the estimation error at each round $t$ for all the 100 experiments. We also plot the average estimation error of the 100 repetitions. (c) We randomly pick one repetition. The loss over round is indicated by the dot. The average cumulative loss is indicated by the line. The dotted line is the reference line indicating the variance of the noise. Here, $\mathbf{E}[\epsilon^T \epsilon] = 0.1667$.

Suppose the transportation cost on edge $(2,3)$ and $(2,5)$ are unknown, and the learner seeks to learn them given the (demand,noisy decision) pairs that arrive sequentially in $T = 1000$ rounds. We assume that $c_e$ for $e \in E$ is within $[1, 10]$. The learning rate is set to $\eta_t = 2/\sqrt{t}$. Then, we implement Algorithm 1 with Cold-start. Figure 4b shows the estimation error of $c$ in each round over the 100 repetitions. We also plot the average estimation error of the 100 repetitions. As shown in this figure, $c_t$ asymptotically converges to the true transportation cost $c_{ture}$ pretty fast. Also. estimation errors over rounds for different repetitions concentrate around the average, indicating that our algorithm is pretty robust to noises. We then randomly pick one repetition and plot the loss over round and the average cumulative loss in Figure 4c. Note that the variance of the noise $\mathbf{E}[\epsilon^T \epsilon] = 0.1667$. We can see that the average cumulative loss asymptotically converges to the variance of the noise.

# 5   Conclusions and final remarks

In this paper, an online learning method to infer preferences or restrictions from noisy observations is developed and implemented. We prove a regret bound for the implicit online learning algorithm under certain regularity conditions, and show that the algorithm is statistically consistent, which guarantees that our algorithm will asymptotically achieves the best prediction error permitted by the inverse model. Experiment results show that our algorithm can learn the parameters with great accuracy, is robust to noises even if some assumptions are not satisfied or difficult to be verified, and achieves a dramatic improvement over the batch learning approach on computational efficacy. Future research directions include the algorithm development with more sophisticated online learning techniques for a stronger performance, and the theoretical investigation with less restriction assumptions and a broader applicability.

### Acknowledgments

This work was partially supported by CMMI-1642514 from the National Science Foundation. This work used the Bridges system, which is supported by NSF award number ACI-1445606, at the Pittsburgh Supercomputing Center (PSC).

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
