[Supplementary Material]

# Supplementary Material for Generalized Inverse Optimization through Online Learning

Chaosheng Dong, Yiran Chen, Bo Zeng

## A  Omitted mathematical reformulations

### A.1  Single level reformulation for the Inverse Linear Optimization Problem

When the objective function is linear, namely, the optimization problem has the following form

$$\min_{\mathbf{x}\in\mathbb{R}^n_+} \quad \mathbf{c}^T\mathbf{x}$$

$$s.t. \quad A\mathbf{x} \geq \mathbf{b}. \tag{LP}$$

Suppose that the right hand side $\mathbf{b}$ changes over time $t$. That is, $\mathbf{b} = \mathbf{b}_t$ at time $t$. When trying to learn $\mathbf{c}$, the single level reformulation the inverse problem is

$$
\begin{aligned}
\min_{\mathbf{c}\in\Theta} \quad & \tfrac{1}{2}\|\mathbf{c} - \mathbf{c}_t\|_2^2 + \eta_t\|\mathbf{y}_t - \mathbf{x}\|_2^2 \\
s.t. \quad & A\mathbf{x} \geq \mathbf{b}_t,\ \mathbf{x} \geq \mathbf{0}, \\
& \mathbf{A}^T\mathbf{u} \leq \mathbf{c}, \\
& \mathbf{x} \leq M_1\mathbf{z}_1, \\
& \mathbf{c} - \mathbf{A}^T\mathbf{u} \leq M_1(1 - \mathbf{z}_1), \\
& \mathbf{u} \leq M_2\mathbf{z}_2, \\
& A\mathbf{x} - \mathbf{b}_t \leq M_2(1 - \mathbf{z}_2), \\
& \mathbf{x} \in \mathbb{R}^n_+,\ \mathbf{u} \in \mathbb{R}^m_+,\ \mathbf{z}_1 \in \{0,1\}^n,\ \mathbf{z}_2 \in \{0,1\}^m,
\end{aligned}
$$

where $M_1$ and $M_2$ are appropriate numbers used to bound $\mathbf{x}$ and $\mathbf{c} - \mathbf{A}^T\mathbf{u}$, $\mathbf{u}$ and $A\mathbf{x} - \mathbf{b}_t$ respectively.

We have a similar single level reformulation when learning the Right-hand side $\mathbf{b}$. Clearly, this is a Mixed Integer Second Order Cone program(MISOCP) when learning either $\mathbf{c}$ or $\mathbf{b}$.

### A.2  Single level reformulation for the Inverse Quadratic Optimization Problem

When the objective functions are quadratic, namely, the optimization problem has the following form

$$\min_{\mathbf{x}\in\mathbb{R}^n} \quad \tfrac{1}{2}\mathbf{x}^T Q\mathbf{x} + \mathbf{c}^T\mathbf{x}$$

$$s.t. \quad A\mathbf{x} \geq \mathbf{b}. \tag{QP}$$

Suppose that $\mathbf{c}$ changes over time $t$. That is, $\mathbf{c} = \mathbf{c}_t$ at time $t$. When trying to learn $\mathbf{b}$, the single level reformulation for the inverse problem is

$$
\begin{aligned}
\min_{\mathbf{b}\in\Theta} \quad & \tfrac{1}{2}\|\mathbf{b} - \mathbf{b}_t\|_2^2 + \eta_t\|\mathbf{y}_t - \mathbf{x}\|_2^2 \\
s.t. \quad & A\mathbf{x} \geq \mathbf{b}, \\
& \mathbf{u} \leq M\mathbf{z}, \\
& A\mathbf{x} - \mathbf{b} \leq M(1 - \mathbf{z}), \\
& Q\mathbf{x} + \mathbf{c}_t - \mathbf{A}^T\mathbf{u} = 0, \\
& \mathbf{b} \in \mathbb{R}^m,\ \mathbf{x} \in \mathbb{R}^n,\ \mathbf{u} \in \mathbb{R}^m_+,\ \mathbf{z} \in \{0,1\}^m,
\end{aligned}
$$

where $M$ is an appropriate number used to bound $\mathbf{u}$ and $A\mathbf{x} - \mathbf{b}$.

We have a similar single level reformulation when learning the objective $\mathbf{c}$. Clearly, this is a Mixed Integer Second Order Cone program(MISOCP) when learning either $\mathbf{c}$ or $\mathbf{b}$.

# B  Omitted Proofs

## B.1  Proof of Lemma 3.1

*Proof.* By Assumption 3.1(b), we know that $S(u, \theta)$ is a single-valued set for each $u \in \mathcal{U}$.
$\forall \mathbf{y} \in \mathcal{Y}, \forall u \in \mathcal{U}, \forall \theta_1, \theta_2 \in \Theta$, without of loss of generality, let $l(\mathbf{y}, u, \theta_1) \geq l(\mathbf{y}, u, \theta_2)$. Then,

$$
\begin{aligned}
|l(\mathbf{y}, u, \theta_1) - l(\mathbf{y}, u, \theta_2)| &= l(\mathbf{y}, u, \theta_1) - l(\mathbf{y}, u, \theta_2) \\
&= \|\mathbf{y} - S(u, \theta_1)\|_2^2 - \|\mathbf{y} - S(u, \theta_2)\|_2^2 \\
&= \langle S(u, \theta_2) - S(u, \theta_1), 2\mathbf{y} - S(u, \theta_1) - S(u, \theta_2) \rangle \\
&\leq 2(B + R)\|S(u, \theta_2) - S(u, \theta_1)\|_2.
\end{aligned}
\tag{1}
$$

The last inequality is due to Cauchy-Schwartz inequality and the Assumptions 3.1(a), that is

$$
\|2\mathbf{y} - S(u, \theta_1) - S(u, \theta_2)\|_2 \leq 2(B + R).
\tag{2}
$$

Next, we will apply Proposition 6.1 in Bonnans and Shapiro [1998] to bound $\|S(u, \theta_2) - S(u, \theta_1)\|_2$.
Under Assumptions 3.1 - 3.2, the conditions of Proposition 6.1 in Bonnans and Shapiro [1998] are satisfied. Therefore,

$$
\|S(u, \theta_2) - S(u, \theta_1)\|_2 \leq \frac{2\kappa}{\lambda}\|\theta_1 - \theta_2\|_2.
\tag{3}
$$

Plugging (2) and (3) in (1) yields the claim. □

## B.2  Proof of Theorem 3.2

*Proof.* we will use Theorem 3.2 in Kulis and Bartlett [2010] to prove our theorem.
Let $G_t(\theta) = \frac{1}{2}\|\theta - \theta_t\|_2^2 + \eta_t l(\mathbf{y}_t, u_t, \theta)$.
We will now show the loss function is convex. The first step is to show that if Assumption 3.3 holds, then the loss function $l(\mathbf{y}, u, \theta)$ is convex in $\theta$. $\forall \mathbf{y} \in \mathcal{Y}, \forall u \in \mathcal{U}, \forall \theta_1, \theta_2 \in \Theta$, we have

$$
\begin{aligned}
&\alpha l(\mathbf{y}, u, \theta_1) + \beta l(\mathbf{y}, u, \theta_2) - l(\mathbf{y}, u, \alpha\theta_1 + \beta\theta_2) \\
=\ & \alpha\|\mathbf{y} - S(u, \theta_1)\|_2^2 + \beta\|\mathbf{y} - S(u, \theta_2)\|_2^2 - \|\mathbf{y} - S(u, \alpha\theta_1 + \beta\theta_2)\|_2^2 \\
=\ & \alpha\|\mathbf{y} - S(u, \theta_1)\|_2^2 + \beta\|\mathbf{y} - S(u, \theta_2)\|_2^2 - \|\mathbf{y} - \alpha S(u, \theta_1) - \beta S(u, \theta_2)\|_2^2 \\
&+ \|\mathbf{y} - \alpha S(u, \theta_1) - \beta S(u, \theta_2)\|_2^2 - \|\mathbf{y} - S(u, \alpha\theta_1 + \beta\theta_2)\|_2^2 \\
=\ & \alpha\beta\|S(u, \theta_1) - S(u, \theta_2)\|_2^2 + \|\mathbf{y} - \alpha S(u, \theta_1) - \beta S(u, \theta_2)\|_2^2 - \|\mathbf{y} - S(u, \alpha\theta_1 + \beta\theta_2)\|_2^2 \\
=\ & \alpha\beta\|S(u, \theta_1) - S(u, \theta_2)\|_2^2 \\
&- \langle \alpha S(u, \theta_1) + \beta S(u, \theta_2) - S(u, \alpha\theta_1 + \beta\theta_2), 2\mathbf{y} - S(u, \alpha\theta_1 + \beta\theta_2) - \alpha S(u, \theta_1) - \beta S(u, \theta_2) \rangle \\
\geq\ & \alpha\beta\|S(u, \theta_1) - S(u, \theta_2)\|_2^2 - \|\alpha S(u, \theta_1) + \beta S(u, \theta_2) - S(u, \alpha\theta_1 + \beta\theta_2)\|_2 \|2\mathbf{y} - S(u, \alpha\theta_1 \\
&+ \beta\theta_2) - \alpha S(u, \theta_1) - \beta S(u, \theta_2)\|_2.
\end{aligned}
\tag{4}
$$

The last inequality is by Cauchy-Schwartz inequality. Note that

$$
\begin{aligned}
&\|\alpha S(u, \theta_1) + \beta S(u, \theta_2) - S(u, \alpha\theta_1 + \beta\theta_2)\|_2 \|2\mathbf{y} - S(u, \alpha\theta_1 + \beta\theta_2) - \alpha S(u, \theta_1) - \beta S(u, \theta_2)\|_2 \\
&\leq 2(B + R)\|\alpha S(u, \theta_1) + \beta S(u, \theta_2) - S(u, \alpha\theta_1 + \beta\theta_2)\|_2 \\
&\leq \alpha\beta\|S(u, \theta_1) - S(u, \theta_2)\|_2 \qquad\qquad \text{(By Assumption 3.3).}
\end{aligned}
\tag{5}
$$

Plugging (5) in (4) yields the result.
Using Theorem 3.2 in Kulis and Bartlett [2010], for $\alpha_t \leq \frac{G_t(\theta_{t+1})}{G_t(\theta_t)}$, we have

$$
R_T \leq \sum_{t=1}^{T} \frac{1}{\eta_t}(1 - \alpha_t)\eta_t l(\mathbf{y}_t, u_t, \theta_t) + \frac{1}{2\eta_t}(\|\theta_t - \theta^*\|_2^2 - \|\theta_{t+1} - \theta^*\|_2^2).
\tag{6}
$$

Notice that

$$
\begin{aligned}
G_t(\theta_t) - G_t(\theta_{t+1}) \quad &= \eta_t(l(\mathbf{y}_t, u_t, \theta_t) - l(\mathbf{y}_t, u_t, \theta_{t+1})) - \tfrac{1}{2}\|\theta_t - \theta_{t+1}\|_2^2 \\
&\leq \tfrac{4(B+R)\kappa\eta_t}{\lambda}\|\theta_t - \theta_{t+1}\|_2 - \tfrac{1}{2}\|\theta_t - \theta_{t+1}\|_2^2 \\
&\leq \tfrac{8(B+R)^2\kappa^2\eta_t^2}{\lambda^2}.
\end{aligned}
\tag{7}
$$

The first inequality follows by applying Lemma 3.1.

Let $\alpha_t = \frac{G_t(\theta_{t+1})}{G_t(\theta_t)}$. Using (7), we have

$$
(1 - \alpha_t)\eta_t l(\mathbf{y}_t, u_t, \theta_t) = (1 - \alpha_t)G_t(\theta_t) = G_t(\theta_t) - G_t(\theta_{t+1}) \leq \tfrac{8(B+R)^2\kappa^2\eta_t^2}{\lambda^2}.
\tag{8}
$$

Plug (8) in (6), and note the telescoping sum,

$$
R_T \leq \sum_{t=1}^T \frac{8(B+R)^2\kappa^2\eta_t}{\lambda^2} + \sum_{t=1}^T \frac{1}{2\eta_t}(\|\theta_t - \theta^*\|_2^2 - \|\theta_{t+1} - \theta^*\|_2^2).
$$

Setting $\eta_t = \frac{D\lambda}{2(B+R)\kappa\sqrt{2t}}$, we can upper bound the second summation by $\frac{4\sqrt{2}(B+R)D\kappa}{\lambda}\sqrt{T}$ since $\|\theta_1 - \theta^*\|_2 \leq 2D$, $\sqrt{t} \leq \sqrt{T}$, and then the sum telescopes. The first sum simplifies using $\sum_{t=1}^T \frac{1}{\sqrt{t}} \leq 2\sqrt{T} - 1$ to obtain the result

$$
R_T \leq \frac{8\sqrt{2}(B+R)D\kappa}{\lambda}\sqrt{T}.
$$

Note that choosing $\eta_t = \frac{1}{\sqrt{t}}$ also yields $\mathcal{O}(\sqrt{T})$ regret, but the result above is tighter. $\qquad\square$

## B.3 Proof of Theorem 3.3

*Proof.* Since $f(\mathbf{x}, u, \theta)$ is strongly convex in $\mathbf{x}$ on $\mathbb{R}^n$ by Assumption 3.1, it is also strictly convex in $\mathbf{x}$ on $\mathbb{R}^n$. Then, all the conditions required in Theorem 3. of Aswani et al. [2018] are naturally satisfied under our assumptions. Applying that theorem yields

$$
\frac{1}{T}\sum_{t\in[T]} l(\mathbf{y}_t, u_t, \theta^T) \xrightarrow{p} \mathbb{E}\left[l(\mathbf{y}, u, \theta^*)\right],
\tag{9}
$$

where $\theta^T = \arg\min_{\theta\in\Theta}\{\sum_{t\in[T]} l(\mathbf{y}_t, u_t, \theta)\}$ is the estimation of the parameter in batch setting.

From Theorem 3.2 we have

$$
\frac{1}{T}\sum_{t\in[T]} l(\mathbf{y}_t, u_t, \theta_t) - \frac{1}{T}\sum_{t\in[T]} l(\mathbf{y}_t, u_t, \theta^T) \leq \frac{8\sqrt{2}(B+R)D\kappa}{\lambda\sqrt{T}} \xrightarrow{p} 0.
\tag{10}
$$

Adding (9) and (10) up, we have the risk consistency result

$$
\frac{1}{T}\sum_{t\in[T]} l(\mathbf{y}_t, u_t, \theta_t) \xrightarrow{p} \mathbb{E}\left[l(\mathbf{y}, u, \theta^*)\right].
$$

$\qquad\square$

## B.4 Proof of Corollary 3.3.1

*Proof.* Note that $\forall\theta \in \Theta$,

$$
\mathbb{E}\left[l(\mathbf{y}, u, \theta)\right] = \mathbb{E}\left[\min_{\tilde{\mathbf{x}}\in S(u,\theta)}\|\mathbf{x} + \epsilon - \tilde{\mathbf{x}}\|_2^2\right] = \mathbb{E}\left[\min_{\tilde{\mathbf{x}}\in S(u,\theta)}\|\mathbf{x} - \tilde{\mathbf{x}}\|_2^2\right] + \mathbb{E}[\epsilon^T\epsilon] \geq \mathbb{E}[\epsilon^T\epsilon].
$$

We further notice that $\mathbb{E}\left[\min_{\tilde{\mathbf{x}}\in S(u,\theta_0)}\|\mathbf{x} - \tilde{\mathbf{x}}\|_2^2\right] = 0$, since $\mathbf{x} \in S(u, \theta_0)$. Therefore, we have

$$
\mathbb{E}\left[l(\mathbf{y}, u, \theta^*)\right] = \mathbb{E}\left[l(\mathbf{y}, u, \theta_0)\right] = \mathbb{E}[\epsilon^T\epsilon].
$$

Then, applying Theorem 3.3 yields the result, since we have shown $\mathbb{E}\left[l(\mathbf{y}, u, \theta^*)\right] = \mathbb{E}[\epsilon^T\epsilon]$. $\qquad\square$

## C Omitted Examples

### C.1 Examples for which Assumption 3.3 holds

Consider for example the following quadratic program

$$\min_{\mathbf{x} \in \mathbb{R}^n} \quad \frac{1}{2}\mathbf{x}^T Q \mathbf{x} + (\mathbf{c} + u)^T \mathbf{x}$$

$$s.t. \quad A\mathbf{x} \geq \mathbf{b}.$$

where $Q$ is a positive semidefinite matrix, and $u$ is the external signal.

Suppose that the parameter we seek to learn is $\mathbf{c}$, all the others are given. If for each $u \in \mathcal{U}$, the optimal solution for the above program is in the interior of the feasible region, which essentially occurs when the external signal $u$ does not has a large range for the constrained QP. Then,

$$S(u, \mathbf{c}_1) = -Q^{-1}(\mathbf{c}_1 + u); \quad S(u, \mathbf{c}_2) = -Q^{-1}(\mathbf{c}_2 + u); \quad S(u, \alpha\mathbf{c}_1 + \beta\mathbf{c}_2) = -Q^{-1}(\alpha\mathbf{c}_1 + \beta\mathbf{c}_2 + u);$$

Then, we have

$$0 = \|\alpha S(u, \mathbf{c}_1) + \beta S(u, \mathbf{c}_2) - S(u, \alpha\mathbf{c}_1 + \beta\mathbf{c}_2)\|_2 \leq \alpha\beta\|S(u, \theta_1) - S(u, \theta_2)\|_2/(2(B + R)).$$

## D Data for the applications

### D.1 Data for learning the consumer behavior

Table 1: True **r**

| 1.180 | 1.733 | 1.564 | 0.040 | 2.443 | 1.055 | 4.760 | 5.000 | 1.258 | 4.933 |
|---|---|---|---|---|---|---|---|---|---|

Table 2: True $Q$

| 2.360 | 0 | 0 | 0 | 0 | 0 | 0 | 0 | 0 | 0 |
|---|---|---|---|---|---|---|---|---|---|
| 0 | 3.465 | 0 | 0 | 0 | 0 | 0 | 0 | 0 | 0 |
| 0 | 0 | 3.127 | 0 | 0 | 0 | 0 | 0 | 0 | 0 |
| 0 | 0 | 0 | 0.0791 | 0 | 0 | 0 | 0 | 0 | 0 |
| 0 | 0 | 0 | 0 | 4.886 | 0 | 0 | 0 | 0 | 0 |
| 0 | 0 | 0 | 0 | 0 | 2.110 | 0 | 0 | 0 | 0 |
| 0 | 0 | 0 | 0 | 0 | 0 | 9.519 | 0 | 0 | 0 |
| 0 | 0 | 0 | 0 | 0 | 0 | 0 | 9.999 | 0 | 0 |
| 0 | 0 | 0 | 0 | 0 | 0 | 0 | 0 | 2.517 | 0 |
| 0 | 0 | 0 | 0 | 0 | 0 | 0 | 0 | 0 | 9.867 |

### D.2 Data for learning the transportation cost

We let $\lambda_1 = 2$, $\lambda_2 = 10$, $u_e = 1.3$ for all $e \in E$, $y_1 = 3$ and $y_2 = 1.5$.

Table 3: True transportation cost for each edge

| $c_{13}$ | $c_{14}$ | $c_{23}$ | $c_{25}$ | $c_{34}$ | $c_{35}$ |
|---|---|---|---|---|---|
| 3.124 | 4.119 | 3.814 | 1.071 | 5.398 | 2.899 |