[Reviews · NeurIPS 2018]

Reviewer 1



Post author response: Thanks for the response. That windowing idea sure did not turn out well! Unfortunately, I still feel that my original score represents my assessment of th epaper. The main idea in this paper is to create an online inverse optimizer. The paper also proves that the proposed solution is statistically consistent and has sqrt(T) regret. The paper then examines the algorithm in the context of utility, budget, and transportation problems. Quality: I rated the work as just above the threshold. it solves a useful problem in an interesting way, but I wish other points of comparison were used. Could a windowing approach be applied to the batch approach or perhaps some approximation to (1)? Also, the key algorithmic contribution seems to be in recognizing that the problem can be structured in a way (KKT conditions) to be solved as a mixed integer second order conic program (MISOCP), that seems a little incremental, in terms of the contribution. I do not mean to imply that there is something incorrect about this, it is not low quality! Clarity: I really liked the clear, step by step structure of this paper. I was never confused about what I was reading and why. Originality: It certainly seems original, but it really does depend on the MISOCP solver. Significance: Here too I was a bit disappointed. I understand in principle why this is an important problem, but I did not find the "sketches of representative applications" all that compelling. A real application with real monetary consequences, or data adapted from a real system would have made the paper more compelling. In addition: At the start of section 3.2 you assert that the resultant problem is a MISOCP, Since this point is the key to the solution, it would have been worth stepping through this just a bit, perhaps moving some of the material from the supplemental materials into the body and adding some more to the supplement.

Reviewer 2



This submission is pretty out of my expertise. I just carefully read the intro part and feel it is well written. One question: you show a nice regret upper bound. It will be nicer if you can show some hardness result (lower bound). In that way, the audience knows better how to appreciate the tightness of your results.

Reviewer 3



Inverse optimization deals with finding the underlying objective function of an optimization problem (mainly in a parametric setup) from observation of inputs and correspoding optimal solutions as the data. This can be done in a batch or in an online fashion from the data. The authors provide an online approach for doing this and prove that it can achieve a regret O(1/sqrt(T)) as a function of data size T. 1. I believe the paper is not written very well. For example, the underlying idea of inverse optimization is not clearly explained as in main references such as [1]. Moreover, the problem has been posed in a quite general setup but at the end the authors have focused on the convex and strongly convex setup where they have used the result already in the literature with some minor modifications. 2. the definition of the function h(x,thet_1,theta_2) at end of page 4 is not clear at all. What is the role of the variable "u" and where does it appear? 3. There are some typoes in the text that should be corrected: 3.1. batching learning --> batch learning 3.2. Also. estimation --> Also, estimation 3.3. rational making decison --> rational decision makers 4. the definition of the function h(x,thet_1,theta_2) at end of page 4 is not clear at all. What is the role of the variable "u" and where does it appear? 5. since the authors have focused on the convex case, there was no need to consider S(u, theta) since the optimal solution would be unique thus S(u, \theta) would be a single point. 6. As far as I see, the main novelty of the paper lies in the fact that the authors have notice that the inverse optimization in an online formulation can be posed as an online learning problem with a cost function that relates to the objective of the optimization problem and have posed it in the context of learning with regret. Also, the proved results seem to be a simple extension of the techniques in "Brian Kulis and Peter L. Bartlett. Implicit online learning". However, this is not clearly written and connected in the paper. So, the paper needs to be modified to clearify this connection. 7. In the "Problem Statement" in Section 2, it is necessary to explain that the inverse optimization is considered in a parametric setup where \theta is the parameter of the objective function to be learned. This is not clearly epxlained in the paper.